# AUTOREGRESSIVE SEMANTIC VISUAL RECONSTRUCTION HELPS VLMS UNDERSTAND BETTER

## ABSTRACT

Typical large vision-language models (LVLMs) apply autoregressive supervision solely to textual sequences, without fully incorporating the visual modality into the learning process. This results in three key limitations: (1) an inability to utilize images without accompanying captions, (2) the risk that captions omit critical visual details, and (3) the challenge that certain vision-centric content cannot be adequately conveyed through text. As a result, these models often prioritize vision-to-language alignment while potentially overlooking fine-grained visual information. While some prior works have explored autoregressive image generation, effectively leveraging autoregressive visual supervision to enhance image understanding remains an open challenge. In this paper, we introduce Autoregressive Semantic Visual Reconstruction (ASVR), which enables joint learning of visual and textual modalities within a unified autoregressive framework. We show that autoregressively reconstructing the raw visual appearance of images does not enhance and may even impair multimodal understanding. In contrast, autoregressively reconstructing the semantic content of images consistently improves comprehension. Notably, we find that even when models are given continuous image features as input, they can effectively reconstruct discrete semantic tokens, resulting in stable and consistent improvements across a wide range of multimodal understanding benchmarks. Our approach delivers significant performance gains and scalability across varying data scales, visual input, visual supervision and model architectures. Specifically, ASVR efficiently improves LLaVA-1.5 by over 3% in average scores across 14 multimodal benchmarks.

## 1 INTRODUCTION

The success of large language models (LLMs) has demonstrated the tremendous potential and scalability of the autoregressive (AR) paradigm. In recent years, extending LLMs' powerful capabilities to multimodal understanding through bridge-style architectures, exemplified by LLaVA (Liu et al., 2023b; 2024a;c), have achieved remarkable performance across vision-language tasks (Liu et al., 2023c; Yue et al., 2023; Fu et al., 2024a; Goyal et al., 2017; Li et al., 2023b; Hudson & Manning, 2019a; Kembhavi et al., 2016). These models (Bai et al., 2023b; Wang et al., 2024c; Yao et al., 2024; Chen et al., 2024; Lu et al., 2024; Wu et al., 2024c), typically adopt a simple yet effective learnable projector to align features from a CLIP-based visual encoder into the text embedding space of LLMs.

However, most of the current large vision-language models (LVLMs) (Wang et al., 2024d; Dong et al., 2024; Liu et al., 2024b; Li et al., 2024) supervise only the textual outputs, overlooking the rich visual modality. Specifically, these models are trained to predict the next token in a text response given both the preceding text and associated images. For example, LLaVA-1.5 Liu et al. (2023a) represents a single 336×336 image with 576 visual tokens, yet applies no explicit supervision to the visual content. As a result, while these models are multimodal in form, they remain predominantly language-centric in nature, with insufficient attention paid to the visual modality.

To overcome the lack of explicit visual supervision, traditional LVLMs rely on image-caption pairs to associate visual content with language. However, this approach suffers from three critical limitations, as shown in Figure 1: (1) Although there is a vast amount of image data available online, most images

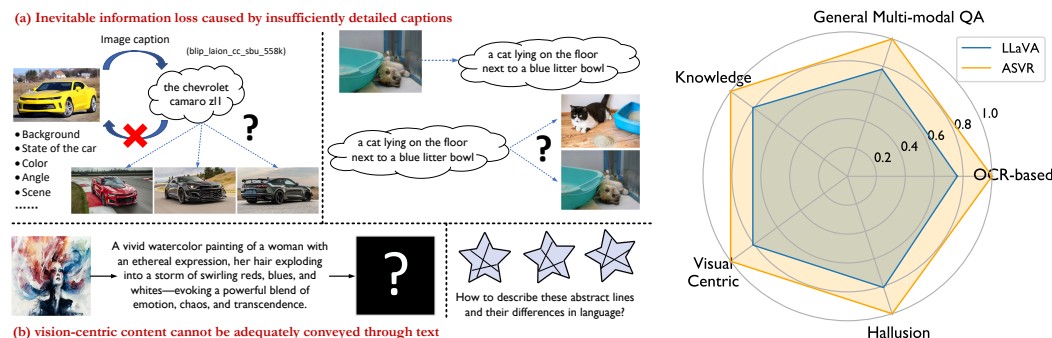

Figure 1: **(Left)** A simple illustration that reflects the information loss faced by language-centric approaches. **(Right)** Our proposed **Autoregressive Semantic Visual Reconstruction (ASVR)** brings significant improvements across various aspects, including General VQA, Visual-centric, Hallucination, and OCR. All the scores are normalized by $x_{\text{norm}} = (x - x_{\min} + 10)/(x_{\max} - x_{\min} + 10)$.

are not accompanied by detailed captions; (2) Even when captions are generated, either manually or by LVLMs, the process is costly, and there remains a risk of omitting critical visual details. The descriptive richness of these captions ultimately constrains the granularity of the model's visual understanding; (3) Some vision-centric content simply cannot be adequately conveyed through text. As the saying goes, "a picture is worth a thousand words", the visual modality serves as an independent and expressive channel that captures spatial relationships, textures, complex compositions, and subtle stylistic cues that text alone struggles to express. In summary, the full spectrum of visual detail in an image is difficult to articulate comprehensively through text, and acquiring large-scale, high-quality, fine-grained captions remains both labor-intensive and expensive.

Recently, several pioneering works have explored unifying visual understanding and generation within the autoregressive paradigm of LLMs (Team, 2024; Wang et al., 2024e; Wu et al., 2024b; Tong et al., 2024b), where visual tokens are supervised through image generation tasks. However, these studies primarily focus on leveraging visual understanding to enhance generation, rather than investigating the reverse direction. Effectively utilizing autoregressive visual supervision to improve visual understanding remains an open challenge. Most recently, Wang et al. (2024b) proposed supervising visual outputs via a denoising approach. However, their method relies on external Diffusion Transformer (DiT) modules for visual supervision and lacks a unified framework that aligns visual and textual modalities under a unified supervision scheme.

In this paper, we introduce **Autoregressive Semantic Visual Reconstruction (ASVR)**, a method that enables joint learning of visual and textual modalities within the unified autoregressive framework of LLMs, without relying on any external modules. Specifically, ASVR allows LVLMs to supervise visual outputs by autoregressively predict the next discrete semantic token of input images, which is prepared by a pretrained semantic visual tokenizer (Song et al., 2025; Wu et al., 2024b; Qu et al., 2024; Xie et al., 2024). Interestingly, we show that autoregressively reconstructing the raw visual appearance of images does not improve and may even degrade multimodal understanding. In contrast, reconstructing semantic visual representation autoregressively consistently enhances the visual understanding capabilities of LVLMs. Notably, we find that even when models are provided with continuous image features as input, they can effectively reconstruct discrete semantic tokens. This setting even outperforms approaches where both input and output use shared discrete semantic visual tokens, resulting in considerable gains, and we also found the unified autoregressive modeling paradigm consistently outperforms its denoising-based counterpart (Wang et al., 2024a).Our approach delivers significant and consistent performance gains across varying data scale settings( LLaVA-1.5-665K (Liu et al., 2023a), LLaVA-Next-779K (Liu et al., 2024b), Bunny-v1_1-data-2M (He et al., 2024)),LLaVA-OV-3.5M (Li et al., 2024) and model architectures such as Vicuna family (Zheng et al., 2023) as well as Mistral (Jiang et al., 2023). Specifically, **ASVR** improves LLaVA-1.5 by 3% in average scores across 14 multimodal benchmarks and the effectiveness is robust across different visual feature types, LLM backbone capacities, data scales, and high-resolution scenarios. These results underscore the importance of explicit semantic visual supervision in training LVLMs. ASVR not only improves visual understanding but also introduces a scalable, unified training strategy, offering a new perspective on autoregressive modeling for multimodal systems.

## 2 RELATED WORK

**Large Vision Language Models** The rapid progress in large language models (LLMs)(Bai et al., 2023a; AI@Meta, 2024; Touvron et al., 2023; Bi et al., 2024; OpenAI, 2023b;a) has showcased their strong generalization and remarkable instruction-following capabilities. To further expand these strengths for interpreting and interacting with the world through both visual and linguistic channels. There has been growing interest in Large Vision-Language Models (LVLMs)(Liu et al., 2023b;a; 2024b), typically trained using a straightforward two-stage visual instruction tuning paradigm (Liu et al., 2023b), and align visual features extracted by visual encoder with the knowledge and reasoning capabilities of LLMs through the lightweight projector. This process involves jointly training the projector and the LLM on visual instruction datasets, with optional fine-tuning of the visual encoder. However, supervision is limited to text outputs. ASVR introduces a novel autoregressive visual semantic supervision mechanism that encourages the LVLM to reconstruct semantic visual tokens, enhancing its multimodal understanding capabilities.

**Visual Autoregression for LVLMs** Some recent approaches (Team, 2024; Qu et al., 2024; Wang et al., 2024e; Wu et al., 2024b;a), introduce autoregressive visual supervision via visual tokenizers, such as VQGAN (Esser et al., 2021) and VQ-VAE (van den Oord et al., 2018), enabling LVLMs to support both multimodal understanding and image generation by predict relevant next visual tokens, which are then decoded into images. In contrast, ASVR focuses specifically on enhancing the multimodal understanding capability of LVLMs. Rather than generating images, ASVR employs autoregressive visual supervision to reconstruct semantic visual tokens within the given continuous image features as input. While prior methods are generative, ASVR adopts the reconstructive approach aimed at promoting perception of visual information.

**Reconstructive Objectives for LVLMs** ROSSWang et al. (2024a) introduced visual supervision for LVLMs by applying denoising objective to reconstruct reconstructs continuous, appearance-level visual features (VAE features). In contrast, ASVR proposes a unified approach by employing autoregressive objective—analogous to that used for text—to reconstruct semantic visual tokens. This design enables seamless integration of visual and textual information under a unified next-token prediction paradigm.

## 3 PRELIMINARIES

**Large Vision Language Models Modeling** To process and represent input sequences from different modalities in a unified manner, Large Vision-Language Models (LVLMs) typically comprise three components: a pre-trained Large Language Model (LLM), a projector commonly implemented as two-layer MLP and a pre-trained visual encoder with semantic aligned.

Given a input RGB image $I \in \mathbb{R}^{H \times W \times 3}$, where $H$ and $W$ denote the image height and width, a pre-trained visual encoder $V_\xi$ is first used to extract image features $\mathbf{z}^I = V_\xi(I)$. These features are then mapped into LLM embedding space via a projector $P_\phi$, producing a sequence of visual features: $\mathbf{H}^I = P_\phi(\mathbf{z}^I) \in \mathbb{R}^{m \times d}$, where $m = h \times w$ denotes the length of visual features, and $d$ is the embedding dimension of LLM. $\xi$ and $\phi$ are the parameters of the visual encoder and projector, respectively. For a textual input $T \in \mathbb{Z}^L$, the LLM's tokenizer is used to produce a sequence of token indices $\mathbf{x}^T = \text{Tokenizer}(T) \in \mathbb{R}^n$. These indices are then transformed into textual embeddings via the LLM's embedding layer $\mathbf{H}^T = \text{Embedding}(x^T) \in \mathbb{R}^{n \times d}$ where $n$ denotes the sequence length.

The final multimodal inputs are formed by concatenating the visual features and textual embeddings, resulting in $[\mathbf{H}^I, \mathbf{H}^T] \in \mathbb{R}^{(m+n) \times d}$, which is then fed into a causal LLM backbone $L_\theta$ with parameters $\theta$ for unified autoregressive modeling:

$$L_\theta([\mathbf{H}^I, \mathbf{H}^T]) = \prod_{i=1}^{n} L_\theta(x_i^T \mid x_{<i}^T, \mathbf{H}^I) \qquad (1)$$

**Training Framework for LVLMs** LVLM training generally involves two stages (Liu et al., 2023b): pre-training and instruction tuning. Pre-training aligns different modalities, enabling the model to jointly understand visual and textual inputs. Instruction tuning further enhances generalization across

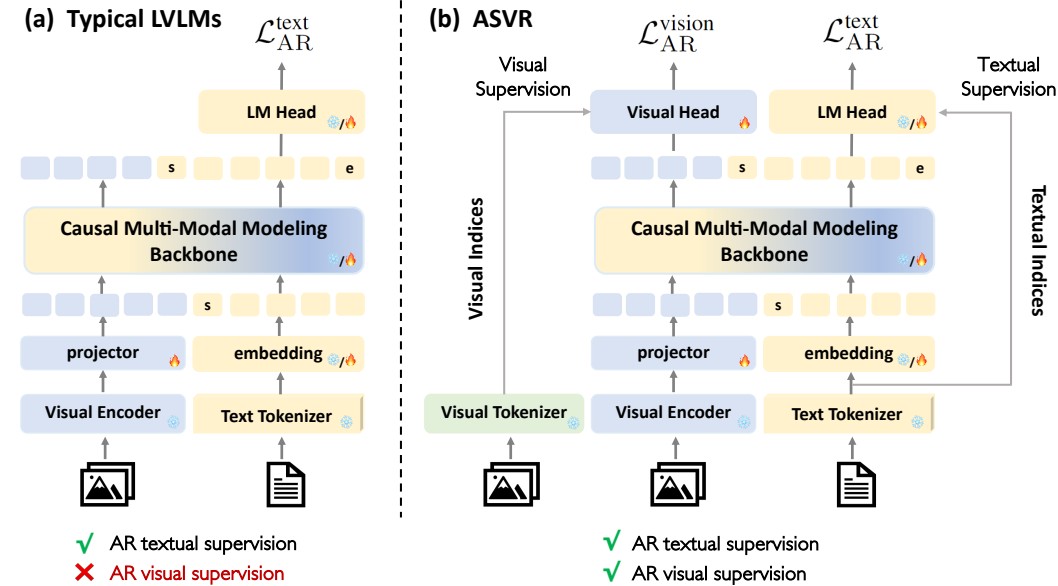

Figure 2: **Left**: the typical LVLM framework exemplified by LLaVA (Liu et al., 2023b). **Right**: overview of **ASVR's** model architecture and training procedure. The input image and its corresponding text are tokenized into sequences of discrete token indices for unified autoregressive supervision over both visual and textual outputs. For each module, the icon before the slash indicates whether it is frozen or tunable during pre-training, while the icon after the slash indicates its configuration during instruction tuning. "s" and "e" denote the start and end of the text tokens, respectively.

diverse downstream tasks such as Visual Question Answering (VQA).The training objective is to maximize the the probability of the target textual responses in autoregressive manner, where only textual responses following the $s$-th token position are supervised.

$$\mathcal{L}_{\mathrm{AR}}^{\text{text}}(\Theta = \{\theta, \xi, \phi\}, T, I) = \frac{-1}{n-s} \sum_{i=s+1}^{n} \log L_\theta(x_i^T \mid x_{<i}^T, \mathbf{H}^I), \quad (2)$$

Here, $\Theta$ denotes the parameters of the entire LVLM. During pre-training, only the parameters of the projector $\phi$ are typically updated, while in instruction tuning, the LLM parameters $\theta$ are also finetuned. The visual encoder $v_\xi$ may either remain frozen (Liu et al., 2023b; Tong et al., 2024a) or be jointly optimized (Li et al., 2024; Dong et al., 2024; Wang et al., 2024d; Liu et al., 2024b).

## 4 METHOD

In this section, we introduce **ASVR**. An overview of the method is provided in Section 4.1, followed by detailed analyses of the visual tokenizer and visual encoder in Sections 4.2 and 4.3, respectively. The training procedure is detailed in Section 4.4. A detailed comparison between the typical LVLMs (LLaVA) and our ASVR is illustrated in Figure 2, highlighting the key innovation of incorporating autoregressive visual supervision to enhance the model's multimodal understanding capabilities.

### 4.1 OVERVIEW

We incorporate autoregressive visual supervision into the typical LVLM's framework described in Section 3 by extending the next-token prediction paradigm to reconstruct and perceive visual inputs. This unified formulation enables the model to seamlessly integrate visual and textual information—first perceiving, then reasoning—thereby establishing a perceptual foundation for image understanding, alleviating the information loss caused by text-only supervision, and ultimately enhancing the LVLM's multimodal understanding capabilities.

As illustrated in Figure 2 (b), we employ the visual tokenizer to convert the input image into discrete sequence of visual token indices, serving as visual supervision signals $\mathbf{x}^I = \text{Tokenizer\_img}(I) \in \mathbb{R}^m$ where $m$ matches the length of the visual features sequence $\mathbf{H}^I$ extracted from pre-trained visual encoder and fed into the LLM backbone. The visual head tailored to the visual tokenizer is then trained to predict the next visual token in autoregressive manner, analogous to textual supervision:

$$\mathcal{L}_{\text{AR}}^{\text{vision}}(\Theta = \{\theta, \xi, \phi\}, I) = \frac{-1}{m} \sum_{i=1}^{m} \log L_\theta(x_i^I \mid x_{<i}^I), \tag{3}$$

Then our final **training objective** is combined with $\mathcal{L}_{\text{AR}}^{\text{vision}}$ and $\mathcal{L}_{\text{AR}}^{\text{text}}$, formulated as

$$\mathcal{L}_{\text{AR}}(\Theta = \{\theta, \xi, \phi\}, I, T) = \mathcal{L}_{\text{AR}}^{\text{vision}} + \mathcal{L}_{\text{AR}}^{\text{text}} \tag{4}$$

This design unifies the learning paradigm across modalities, enabling joint optimization of both vision and language under shared autoregressive objective. Importantly, it also compels the model to first develop coherent visual sensor, which subsequently serves as foundation for more accurate and contextually grounded multimoda understanding.

## 4.2 VISUAL TOKENIZER

Visual tokenizer convert input images into one-dimensional sequences of discrete visual codes through vector quantization(VQ) by learning a fixed-size visual codebook, then look up the corresponding features by codes into the codebook as inputs to the LMM. Additionally, the visual tokenizer defines visual supervision targets by determining the granularity and representations of the discrete visual token indices, which play a critical role in the visual reconstruction and perception. There are two type of visual tokenizer.

**Visual Appearance Tokenizer** A visual appearance tokenizer (Esser et al., 2021; Team, 2024) is optimized with the objective of reconstructing the input image, where utilize reconstruction loss typically combining pixel-wise L2 loss (Dosovitskiy & Brox, 2016), LPIPS lossZhang et al. (2018) and adversarial loss Isola et al. (2017) for reconstruction ability. The resulting sequence of token indices represents a quantized mapping of the image's pixel-level features. Using Pixel-based tokenizer to provide visual pixel supervision targets will guide the LVLM to focus on low-level pixel feature reconstruction and perception.

**Visual Semantic Tokenizer** A visual semantic tokenizer (Qu et al., 2024; Wu et al., 2024b; Xie et al., 2024; Song et al., 2025) is is trained to align image features with textual semantics, typically using a contrastive loss (Radford et al., 2021) to enhance cross-modal alignment. The resulting sequence of token indices represents a quantized mapping of the image's high-level semantic features. Using Semantic-based tokenizer to provide semantic visual supervision targets will guide the LVLM to focus on semantically meaningful aspects reconstruction and perception of the image, thereby promoting more effective multimodal understanding.

## 4.3 VISUAL ENCODER

The visual encoder provides continuous visual features as inputs to the LMM, directly influencing the effectiveness of visual information modeling. To enhance multimodal understanding, it is crucial to employ a visual encoder that is semantically aligned with textual representations (Wu et al., 2024b; Qu et al., 2024; Wu et al., 2024a), thus enabling the extraction of high-level, semantically meaningful image features. Typically, such visual encoders adopt transformer-based (Dosovitskiy et al., 2021) architecture, trained using contrastive loss (Radford et al., 2021) to align closely with textual semantics and directly convert input images into one-dimensional sequences of continuous feature vectors.

## 4.4 TRAINING RECIPE

As shown in Figure 2, we visualize our training recipe, which extends the standard LVLM training framework by incorporating visual supervision to enable unified autoregressive modeling over both

visual inputs and textual responses. Specifically, during the pre-training stage, we focus solely on optimizing the projector and the visual head. This stage aligns visual representations sequence with the LVLM's semantic space, allowing the model to develop an initial perception of image features by learning the mapping between continuous visual features and discrete visual token indices. In the instruction tuning stage, we further fine-tune the parameters of the LLM backbone. Leveraging diverse vision-language instruction data, the model is guided to perform deeper semantic sensing of visual content, thereby enhancing its ability to understand and reason across modalities in a more comprehensive manner.

## 5 EXPERIMENTS

In this section, we present a comprehensive set of controlled experiments to evaluate the effectiveness of our method **(ASVR)** within typical LVLM's frameworks (Liu et al., 2023b) across a diverse range of multimoda understanding tasks.We begin by detailing our experimental setup. Then, we analyze the impact of different visual encoders and visual tokenizers on the model's performance. Finally, we further validate the generalization and adaptability of our method across various LLM backbones with different parameter scales and under varying amounts of training data.

### 5.1 EXPERIMENTAL SETUP

**Implementation Details.** We implement our experiments baseline on the LLaVA-1.5 (Liu et al., 2023a) settings only with textual supervision detaily discussed in sec 3. We utilize Vicuna-1.5-7B (Zheng et al., 2023) as the LLM backbone and initialize visual encoder with the pretrained weights from SigLIP-SO400M-patch14-384 (Alabdulmohsin et al., 2023) to support continuous visual features for LMM. For visual tokenizer, we employ both visual appearance tokenizer and visual semantic tokenizer proposed in DualToken (Song et al., 2025) to construct visual supervision targets, which convert input images into $27 \times 27 \times 8$ visual semantic or appearance token sequences, with a residual depth of $D = 8$. The visual head also derived from DualToken, is integrated and aligned with the chosen visual tokenizer to ensure architectural compatibility. Additional training details and architecture of visual head are provided in Appendix. The training data is LLaVA-558K (Liu et al., 2023b) and LLaVA-1.5-665K (Liu et al., 2023b) for the pre-training stage and the instruction tuning stage, respectively.

**Evaluation Details** We conduct a comprehensive evaluation of model's capabilities on 14 widely used vision-language understanding benchmarks. Specifically, the general multimodal benchmarks include MMBench (Liu et al., 2024d) English dev split(MMB), GQA (Hudson & Manning, 2019b), SEED-Image(SEED) (Li et al., 2023a) and MME sum (Fu et al., 2024b). For OCR-based question answering, we assessed performance on TextVQA(TVQA) (Singh et al., 2019), ChartQA(CQA) (Masry et al., 2022), DocVQA(DVQA) (Mathew et al., 2021) and OCRBench(OCRB) (Liu et al., 2024e) . For knowledge-based question answering, we utilize MMMU validation split (Yue et al., 2024), AI2D (Kembhavi et al., 2016). Additionally, we evaluated hallucination robustness on POPE (Li et al., 2023c), Hallusionbench(Hbench) (Guan et al., 2024) and visual-centric tasks on MMVP (Tong et al., 2024c) and RealworldQA(RQA) (xAI, 2024). Evaluation prompts can be found in Appendix.

### 5.2 MAIN RESULTS

**The Effectiveness of ASVR** As shown in Table 1, with the configuration of the continuous-based visual encoder (SigLIP), we observe ASVR consistent and significant performance improvements across all 14 benchmarks, increasing the average score from **46.8** to **49.8**, with 3%. Notably, the gains are evident even on knowledge-based QA such as MMMU (Yue et al., 2024) and AI2D (Kembhavi et al., 2016), suggesting that reconstructing and and perceiving visual inputs can enhance the model's cognitive reasoning abilities. Furthermore, substantial improvements are also observed on fine-grained tasks such as OCRBench (Liu et al., 2024e), MMVP (Tong et al., 2024c), and HallusionBench (Guan et al., 2024). In particular, HallusionBench sees an increase of nearly 10 points, further validating the effectiveness of our method. Moreover, under the configuration with a discrete-based visual encoder (DualToken), semantic visual supervision also yields notable performance gains over the baseline. This further demonstrates the generalizability and robustness of our method.

Table 1: **The impact of ASVR under different combinations of visual tokenizers and visual encoders across multimoda understanding benchmarks.** "✗" indicates the use of textual supervision only, while "✓" denotes the inclusion of visual supervision by computing additional $\mathcal{L}_{AR}^{vision}$. "Sem." refers to using visual semantic tokenizer to construct visual supervision targets; "App." denotes visual appearance tokenizer; "App.+Sem." represents dual supervision, where both visual semantic and visual appearance tokenizers are used independently to compute their respective $\mathcal{L}_{AR}^{vision}$, which are then summed. ASVR utilize Semantic Supervision

| | $\mathcal{L}_{AR}^{vision}$ | Visual Tokenizer | OCR | | | | General | | | | Knowledge | | Visual-Centric | | Hallusion | | AVG |
|---|---|---|---|---|---|---|---|---|---|---|---|---|---|---|---|---|---|
| | | | TVQA | DVQA | OCRB | CQA | MMB | MME | SEED | GQA | MMMU | AI2D | RQA | MMVP | Hbench | POPE | |
| | | | | | | Dualtoken (Discrete Visual Features) | | | | | | | | | | | |
| LLaVA | ✗ | - | 49.3 | 20.0 | 29.5 | 12.4 | 60.4 | 56.9 | 63.1 | 56.2 | 31.2 | 50.4 | 50.2 | 24.7 | 21.8 | 80.7 | 43.3 |
| **ASVR** | ✓ | Sem. | 55.5(+6.2) | 21.4(+1.4) | 32.4(+2.9) | 14.7(+2.3) | 62.3(+1.9) | 57.7(+0.8) | 65.4(+2.3) | 57.1(+0.9) | 32.0(+0.8) | 53.5(+3.1) | 52.3(+2.1) | 26.0(+1.3) | 27.7(+5.9) | 76.8(-3.9) | 45.3 |
| | | | | | | SigLIP-ViT-SO400M/14@384 (Continuous Visual Features) | | | | | | | | | | | |
| LLaVA | ✗ | - | 56.0 | 21.1 | 31.3 | 14.6 | 64.0 | 67.2 | 63.8 | 60.5 | 32.7 | 53.5 | 52.0 | 28.7 | 23.9 | 85.9 | 46.8 |
| Appearance Supervise | ✓ | App. | 53.7(-2.3) | 17.8(-3.3) | 30.2(-1.1) | 14.4(-0.2) | 61.6(-2.4) | 68.7(-1.5) | 59.5(-4.3) | 57.8(-2.7) | 33.1(+0.4) | 53.7(+0.2) | 49.3(-2.7) | 22.0(-6.7) | 24.0(+0.1) | 84.1(-1.8) | 45.0 |
| Dual Supervise | ✓ | App.+Sem. | 59.4(+3.4) | 23.7(+2.6) | 33.5(+2.2) | 16.1(+1.5) | 65.6(+1.6) | 70.2(+3.0) | 66.1(+2.3) | 61.5(+1.0) | 34.0(+1.3) | 56.3(+2.8) | 53.5(+1.5) | 22.0(-6.7) | 30.7(+6.8) | 86.3(+0.4) | 48.5 |
| **ASVR** | ✓ | Sem. | **59.5**(+3.5) | **24.3**(+3.2) | **35.4**(+4.1) | **16.4**(+1.8) | **66.1**(+2.1) | **72.8**(+5.6) | **66.4**(+2.6) | **61.5**(+1.0) | 33.9(+1.2) | **57.0**(+3.5) | **54.1**(+2.1) | **30.0**(+1.3) | **33.7**(+9.8) | **86.3**(+0.4) | **49.8** |

**Semantic v.s. Appearance** Specifically, ASVR incorporating semantic supervision alone yields the highest average performance across benchmarks, outperforming even the dual supervision setting that combines both appearance and semantic visual indices. In contrast, applying appearance-only supervision degrades model performance compared to the baseline. These results highlight that guiding the LVLM to reconstruct and perceive high-level semantic visual information of the input image, rather than low-level appearance details, more effectively enhances its multimoda understanding capabilities.

**Continuous vs. Discrete** We adopt SigLIP-ViT-SO400M/14@384 (Zhai et al., 2023) to provide continuous visual features, while employing visual semantic tokenizer from Dualtoken (Song et al., 2025) to generate discrete visual features; both approaches aligned with textual semantics. Our experimental results indicate that, regardless of whether autoregressive semantic visual supervision is applied, the configuration of using continuous visual features consistently outperforms its discrete features counterpart across all benchmarks. This performance gap may be attributed to image feature degradation introduced by vector quantization in discrete encoding, which can lead to loss of fine-grained visual information crucial for downstream multimoda understanding.

**Discussion** The combination of visual encoder for provide visual features and visual semantic tokenizer for constructing semantic visual supervision targets proves to the most effective model configuration. The visual encoder avoids the visual information loss typically introduced by vector quantization, thereby providing better visual inputs for the LMM. Meanwhile, semantic supervision guides the LVLM reconstruct high-level, semantically meaningful aspects of the image, which are benefit for multimoda understanding.Notably, our findings demonstrate that continuous visual inputs with discrete semantic visual supervision targets can be seamlessly integrated into the unified autoregressive next-token prediction paradigm in the same manner as language. This formulation enables the LVLM to reconstruct and perceive visual semantic information, enhancing LVLM's capacity for comprehensive multimoda understanding. We further demonstrate that the unified autoregressive modeling paradigm consistently surpasses its denoising-based counterpart (Wang et al., 2024a), with results provided in the Appendix A.6.

## 5.3 METHOD GENERALITY

We validate the generalization and robustness of ASVR in enhancing multimodal understanding under different data scales and diverse LLM backbone configurations, as summarized in Table 2.

**The Impact of Data Scaling** To investigate the effect of training data scale, we also evaluate ASVR under larger training data. we adopt Bunny-pretrain-LAION-2M(He et al., 2024) for pre-training and Bunny-v1_1-data-2M(He et al., 2024) for instruction tuning. We compare the performance of ASVR against the baseline across different data scales to assess its robustness and effectiveness. As

Table 2: **The Generality of ASVR under different training data scale and LLM backbone across multimoda understanding benchmarks.** "✗" indicates the use of textual supervision only, while "✓" denotes the inclusion of semantic visual supervision by computing additional $\mathcal{L}_{\mathrm{AR}}^{\mathrm{vision}}$. Visual encoder(SigLIP-ViT-SO400M/14@384) are both utilized for ASVR and baseline. "/" separates the data scale used for pre-training (left) and instruction tuning (right).

| | $\mathcal{L}_{\mathrm{AR}}^{\mathrm{vision}}$ | LLM backbone | Data Scale | OCR | | | | General | | | | Knowledge | | Visual-centric | | Hallusion | | AVG |
|---|---|---|---|---|---|---|---|---|---|---|---|---|---|---|---|---|---|---|
| | | | | TVQA | DVQA | OCRB | CQA | MMB | MME | SEED | GQA | MMMU | AI2D | RQA | MMVP | Hbench | POPE | |
| | | | | | | | | With Different Data Scale | | | | | | | | | | |
| LLaVA | ✗ | Vicuna-1.5-7B | 2M/2M | 61.6 | 43.8 | 35.4 | 38.7 | 68.4 | 74.9 | 67.9 | 61.7 | 40.6 | 64.6 | 56.1 | 34.8 | 36.9 | 85.6 | 55.1 |
| ASVR | ✓ | Vicuna-1.5-7B | 2M/2M | 60.6(-1.0) | 43.1(-0.7) | 36.2(+0.8) | 38.9(+0.2) | 68.6(+0.2) | 76.2(+1.3) | 68.7(+0.8) | 62.0(+0.3) | 41.4(+0.8) | 64.8(+0.2) | 55.9(-0.2) | 35.9(+1.1) | 42.2(+5.3) | 85.7(+0.1) | 55.7 |
| | | | | | | | | With Different LLM Backbone | | | | | | | | | | |
| LLaVA | ✗ | Mistral-7B | 558K/665K | 50.8 | 15.7 | 34.6 | 15.2 | 65.9 | 66.9 | 67.9 | 62.4 | 32.0 | 53.0 | 55.0 | 35.3 | 32.7 | 86.6 | 48.1 |
| ASVR | ✓ | Mistral-7B | 558K/665k | 54.9(-4.1) | 17.9(+2.2) | 34.1(-0.5) | 15.6(+0.4) | 67.1(+1.2) | 71.5(+4.6) | 68.3(+0.4) | 62.5(+0.1) | 32.6(+0.6) | 54.5(+1.5) | 55.4(+0.4) | 35.7(+0.4) | 35.0(+2.3) | 86.8(+0.2) | 49.4 |

shown in Table 1 and Table 2, ASVR consistently yields substantial improvements over the baseline across different training data scales. Furthermore, as the amount of training data increases, overall model performance improves. However, ASVR maintains a consistent performance margin over the baseline, demonstrating its ability to more effectively leverage additional data through autoregressive semantic visual reconstruction, we also show results on larger and more comprehensive datasets, such as LLaVA-OV-3.5M (Li et al., 2024) in Appendix A.4.

**The Impact of LLM Backbone Capacities** We further evaluate the generalization capability of ASVR across different LLM backbones to examine its robustness to variations in backbone capacities and architectures. Specifically, we extend our experiments to Mistral-7B(Jiang et al., 2023), which differs from the LLaMA family (Zheng et al., 2023). This evaluation allows us to rigorously test the flexibility and adaptability of ASVR, assessing its performance when integrated into different LLMs.As shown in Table 2, ASVR consistently surpasses the baseline across a variety of multimodal benchmarks, maintaining strong performance advantages regardless of backbone variations. These results demonstrating both its robustness and adaptability in diverse LLM configurations. The backbone scaling experiment and clear scaling law table will provide in Appendix A.4.

## 5.4 High Resolution Adaptation

ASVR is also compatible with existing high-resolution strategies and can further enhance the multimodal understanding capabilities of LMMs. To evaluate the effectiveness of ASVR under high-resolution configurations, we upscale the input resolution of both ASVR and the baseline models to $1152 \times 1152$, while keeping the training conditions identical. We use LLaVA-558K(Liu et al., 2023b) for the pre-training stage and LLaVA-Next-779K(Liu et al., 2024b) for instruction tuning following LLaVA-Next settings (Liu et al., 2024b).

Table 3: **The High Resolution Adaptation of ASVR across multimoda understanding benchmarks.** "✗" indicates the use of textual supervision only, while "✓" denotes the inclusion of semantic visual supervision by computing additional $\mathcal{L}_{\mathrm{AR}}^{\mathrm{vision}}$. Visual encoder(SigLIP-ViT-SO400M/14@384) and $1152 \times 1152$ input resolution are both utilized for ASVR and baseline."/" separates the data scale used for pre-training (left) and instruction tuning (right).

| | $\mathcal{L}_{\mathrm{AR}}^{\mathrm{vision}}$ | LLM backbone | Data Scale | OCR | | | | General | | | | Knowledge | | Visual-centric | | Hallusion | | AVG |
|---|---|---|---|---|---|---|---|---|---|---|---|---|---|---|---|---|---|---|
| | | | | TVQA | DVQA | OCRB | CQA | MMB | MME | SEED | GQA | MMMU | AI2D | RQA | MMVP | Hbench | POPE | |
| LLaVA | ✗ | Vicuna-v1.5-7B | 558K/779k | 58.1 | 44.1 | 39.5 | 47.5 | 66.6 | 74.1 | 66.8 | 62.0 | 35.8 | 62.8 | 57.8 | 30.0 | 40.6 | 84.5 | 55.0 |
| ASVR | ✓ | Vicuna-v1.5-7B | 558k/779K | 58.9(+0.8) | 48.9(+4.8) | 45.6(+6.1) | 49.3(+1.8) | 68.0(+1.4) | 76.7(+2.6) | 67.2(+0.4) | 62.4(+0.4) | 36.9(+1.1) | 65.4(+2.6) | 57.6(-0.2) | 31.9(+1.9) | 43.7(+3.1) | 86.5(+2.0) | 57.1 |

As shown in Table 3, under high-resolution configurations, ASVR consistently outperforms the baseline by 2% in average scores across 14 multimodal benchmarks, further demonstrating its flexibility and robustness across different input resolutions.

Table 4: **Ablation study for various ASVR configurations.** This table presents a comparison of various ASVR settings, including semantic tokenizer, varied the degree of alignment with text (e.g., DualToken-12M vs. DualToken-3M (Song et al., 2025)), and the training strategy, where "PT/IT" denotes that semantic visual supervision is applied during both the pre-training and instruction tuning stages, while "IT" indicates that semantic visual supervision is applied only during instruction tuning.

| Ablated Aspects | Original | Ablated Setting | OCR | | | | General | | | | Knowledge | | Visual-centric | | Hallucination | | AVG |
|---|---|---|---|---|---|---|---|---|---|---|---|---|---|---|---|---|---|
| | | | TVQA | DVQA | OCRB | CQA | MMB | MME | SEED | GQA | MMMU | AI2D | RQA | MMVP | HBench | POPE | |
| Semantic Tokenizer | DualToken-12M | DualToken-3M | 57.8(-1.7) | 25.4(+1.1) | 33.1(-2.3) | 16.2(-0.2) | 67.2(+1.1) | 70.3(-2.5) | 64.8(-1.6) | 60.0(-1.5) | 31.8(-2.1) | 55.9(-1.1) | 54.3(+0.2) | 24.7(-5.3) | 33.0(-0.7) | 86.1(-0.2) | 48.6 |
| Training Strategy | PT/IT | IT | 55.3(-4.2) | 18.9(-5.4) | 29.5(-5.9) | 14.0(-2.4) | 61.2(-4.9) | 67.8(-5.0) | 60.5(-5.9) | 58.3(-3.2) | 33.4(-0.5) | 52.6(-4.4) | 52.3(-1.8) | 20.8(-9.2) | 30.0(-3.7) | 84.9(-1.4) | 45.7 |
| ASVR | - | - | 59.5 | 24.3 | 35.4 | 16.4 | 66.1 | 72.8 | 66.4 | 61.5 | 33.9 | 57.0 | 54.1 | 30.0 | 33.7 | 86.3 | 49.8 |

## 5.5 ABLATION STUDY

**The Impact of Semantic Tokenizer**    Increasing the degree of alignment with text for semantic tokenizer leads to performance of ASVR. we use different semantic tokenizers to construct semantic visual supervision targets: DualToken-3M, which achieves zero-shot ImageNet classification accuracy of 78.6% (Deng et al., 2009), and DualToken-12M, which achieves 81.6% and thus exhibits stronger semantic alignment. As shown in Table 4, ASVR equipped with the better-aligned DualToken-12M consistently outperforms the variant using DualToken-3M across the majority of multimodal benchmarks, with the average performance improving by more than 2%. These results demonstrate that employing better semantically aligned visual tokenizer provides semantic visual supervision targets with more meaningful aspects of the image, and further support our claim that Semantic Visual Reconstruction plays a key role in enhancing the multimodal understanding capabilities of LVLMs. Moreover, when the supervised visual tokenizer provides richer semantic information, ASVR achieves stronger performance. We present the results obtained using discrete SigLIP2 (Tschannen et al., 2025) as visual tokenizer which contain richer semantic visual information in the Appendix A.5.

**The Impact of Training Strategy**    We explore different training strategies for ASVR, comparing whether to apply semantic visual supervision in both the pre-training and instruction tuning stages, or to apply it only during instruction tuning, while keeping the pre-training stage purely with text-based autoregressive training. As shown in Table 4, incorporating semantic visual supervision to support visual autoregressive training in both the pre-training and instruction tuning stages consistently outperforms the single-stage variant across all benchmarks, achieving an average performance gain of nearly 6%. This further underscores the importance of Semantic Visual Reconstruction during the pre-training phase, as it enables the model to develop a more complete perception of visual information. By doing so, it enhances vision-language alignment and mitigates the information loss associated with relying solely on textual supervision.

## 6 CONCLUSION

In summary, we introduced **Autoregressive Semantic Visual Reconstruction (ASVR)**, enabling joint learning of visual and textual modalities within a unified autoregressive framework and effectively improving multimodal understanding capability of LVLMs. Unlike conventional LVLMs framework, which predominantly rely on textual autoregressive supervision and frequently neglect crucial visual details, ASVR explicitly integrates semantic visual supervision to foster deep perception of visual inputs. Our findings indicate that reconstructing raw visual appearance autoregressively does not benefit, and can even impair multimodal understanding. Conversely, autoregressively reconstructing semantic visual representations of images consistently enhances performance across diverse multi-modal tasks and also outperform its denoising-based counterpart. Remarkably, even with continuous visual features as input, ASVR effectively reconstructs discrete semantic tokens, yielding stable and substantial improvements on various multimodal benchmarks. This effectiveness is robust across different visual feature types, LLM backbone capacities, data scales, and high-resolution scenarios, underscoring ASVR's adaptability, scalability and versatility. Future work aims to incorporate image generation capabilities into ASVR, leveraging unified visual autoregressive supervision to seamlessly integrate understanding and generation, thus broadening applicability across diverse downstream tasks.

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

## A    APPENDIX

### A.1    USE OF LLMS IN IN PAPER WRITING

In preparing this paper, Large Language Models (LLMs) were employed to support the refinement of writing. Their role was limited to enhancing the linguistic presentation of the paper by improving readability, clarity, and stylistic consistency. Specifically, the models were used for tasks such as rephrasing sentences, checking grammar, and streamlining the flow of the text. We emphasize that the LLMs were not involved in generating research ideas, designing methodologies, or conducting experiments. All conceptual development, methodological design, and analytical work were carried out solely by the authors. The contribution of the LLMs was restricted to language-level improvements and did not extend to the scientific substance of the work. The authors retain complete responsibility for the content of this paper, including passages revised with LLM assistance. Care has been taken to ensure that the use of LLMs complies with ethical standards and does not give rise to plagiarism or any form of scientific misconduct.

### A.2    QUALITATIVE COMPARISON

We visualize attention-score maps from several cases, illustrating the attention distribution of the last token with respect to all visual tokens, as shown in Figure 3. Compared to the baseline (LLaVA), our ASVR method consistently demonstrates more precise focus on image regions relevant to the given textual query. This highlights that incorporating semantic visual supervision via the autoregressive semantic visual reconstruction objective $\mathcal{L}_{\text{AR}}^{\text{vision}}$ effectively enhance its ability to accurately associate textual descriptions with corresponding visual elements.

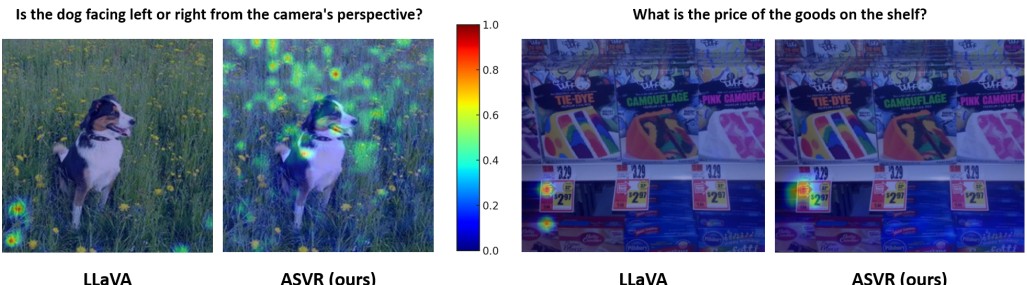

Figure 3: Qualitative comparison on attention maps, where we keep the same LLM and training data. With extra vision-centric supervision signals, ROSS urges the model to focus on specific image contents corresponding to the question with higher attention values.

### A.3    EVALUATION PROMPTS

All prompts used for evaluation benchmarks are released and summarized in Table5 following Cambrian-1 (Tong et al., 2024a).

### A.4    THE SCALABILITY OF ASVR

we show the clear scaling study along two axes in Table 6 and Table 7:

**Data scaling** We train on four datasets—LLaVA-1.5-556K (Liu et al., 2023b), LLaVA-Next-779K (Liu et al., 2024c), Bunny-2M (He et al., 2024), and LLaVA-SI-3.5M (Li et al., 2024) to isolate the effect of data volume.

**Backbone scaling** Using the same Vicuna family, we vary only the parameter count and scale up to a 13B model (the maximum allowed by our computational budget).

Table 5: **Listing the prompts used in the evaluation of each benchmark.**

| Benchmark | Prompt |
|---|---|
| TextVQA (Singh et al., 2019) | Answer the question using a single word or phrase. |
| DocVQA (Mathew et al., 2021) | Answer the question using a single word or phrase. |
| OCRBench (Liu et al., 2024e) | Give the short answer directly. |
| ChartQA (Masry et al., 2022) | Answer the question using a single number or phrase. |
| MMBench (Liu et al., 2024d) | Answer with the option's letter from the given choices directly. |
| MME (Fu et al., 2024b) | Answer the question using a single word or phrase. |
| SEED-Image (Li et al., 2023a) | Answer with the option's letter from the given choices directly. |
| GQA (Hudson & Manning, 2019b) | Answer the question using a single word or phrase. |
| MMMU (Yue et al., 2024) | Answer with the option's letter from the given choices directly. |
| AI2D (Kembhavi et al., 2016) | Answer with the option's letter from the given choices directly. |
| RealworldQA (xAI, 2024) | Please answer directly with only the letter of the correct option and nothing else. |
| MMVP (Tong et al., 2024c) | Answer with the option's letter from the given choices directly. |
| Hallusionbench (Guan et al., 2024) | Answer the question using a single word or phrase. |
| POPE (Li et al., 2023c) | Answer the question using a single word or phrase. |

Table 6: **The scaling relationship between computational cost (FLOPs) and average performance score across different scale dataasets with the same LLM backbone-vicuna-1.5-7B.**

| Data | FLOPs (×1e19) | Avg Score |
|---|---|---|
| LLaVA-1.5-556K (Liu et al., 2023b) | 1.53 | 49.8 |
| LLaVA-Next-779K (Liu et al., 2024c) | 2.49 | 55.1 |
| Bunny-2M (He et al., 2024) | 4.52 | 55.7 |
| LLaVA-SI-3.5M (Li et al., 2024) | 7.51 | 57.9 |

Table 7: **The scaling relationship between computational cost (FLOPs) and average performance score across different scale backbone parameters with the same training dataset-LLaVA-1.5-556k.**

| LLM Backbone | FLOPs (×1e19) | Avg Score |
|---|---|---|
| vicuna-1.5-7B | 1.53 | 49.8 |
| vicuna-1.5-13B | 2.58 | 52.4 |

### A.4.1 THE IMPACT OF BACKBONE SCALING

We further evaluate the generalization capability of **ASVR** under the larger-scale LLM backbone. Specifically, we extend our experiments to Vicuna-v1.5-13B(Zheng et al., 2023), The training data is LLaVA-558K (Liu et al., 2023b) and LLaVA-1.5-665K (Liu et al., 2023b) for the pre-training stage and the instruction tuning stage respectively, keeping the training conditions identical. As shown in Table8, **ASVR** consistently outperforms the baseline across a wide range of multimodal benchmarks, demonstrating its effectiveness in scaling with larger LLM backbones.

Table 8: **The Generality of ASVR with LLM backbone scaling across multimodal understanding benchmarks.** "✗" indicates the use of textual supervision only, while "✓" denotes the inclusion of semantic visual supervision by computing additional $\mathcal{L}_{AR}^{vision}$. Visual encoder(SigLIP-ViT-SO400M/14@384) are both utilized for ASVR and baseline. "/" separates the data scale used for pre-training (left) and instruction tuning (right).

| | $\mathcal{L}_{AR}^{vision}$ | LLM backbone | Data Scale | OCR | | | | General | | | | Knowledge | | Visual-centric | | Hallusion | | AVG |
|---|---|---|---|---|---|---|---|---|---|---|---|---|---|---|---|---|---|---|
| | | | | TVQA | DVQA | OCRB | CQA | MMB | MME | SEED | GQA | MMMU | AI2D | RQA | MMVP | Hbench | POPE | |
| LLaVA | ✗ | Vicuna-v1.5-13B | 558K/665k | 57.2 | 22.1 | 32.4 | 15.1 | 67.1 | 68.9 | 65.6 | 60.4 | 35.6 | 54.9 | 54.8 | 34.0 | 32.9 | 86.8 | 49.1 |
| **ASVR** | ✓ | Vicuna-v1.5-13B | 558k/665K | **61.6**+4.4 | **27.3**+5.2 | **37.1**+4.7 | **18.4**+3.3 | **70.8**+3.7 | **74.9**+6.0 | **68.7**+3.1 | **62.8**+2.4 | **36.4**+0.8 | **60.0**+5.1 | **56.0**+1.2 | **35.3**+1.3 | **36.8**+3.9 | **87.5**+0.7 | **52.4** |

### A.4.2 THE SCALING ON LARGER DATA

We further evaluate the generalization capability of **ASVR** under larger and more comprehensive datasets, LLaVA-OV-3.5M (Li et al., 2024). As shown in Table 9, **ASVR** consistently outperforms the baseline across a wide range of multimodal benchmarks, demonstrating its effectiveness in scaling with larger and more comprehensive datasets.

Table 9: **The Generality of ASVR with larger and more comprehensive datasets LLaVA-OV-3.5M across multimodal understanding benchmarks.** "✗" indicates the use of textual supervision only, while "✓" denotes the inclusion of semantic visual supervision by computing additional $\mathcal{L}_{AR}^{vision}$. Visual encoder(SigLIP-ViT-SO400M/14@384) are both utilized for ASVR and baseline.

| | $\mathcal{L}_{AR}^{vision}$ | LLM backbone | Data | OCR | | | | General | | | | Knowledge | | Visual-centric | | Hallusion | | AVG |
|---|---|---|---|---|---|---|---|---|---|---|---|---|---|---|---|---|---|---|
| | | | | TVQA | DVQA | OCRB | CQA | MMB | MME | SEED | GQA | MMMU | AI2D | RQA | MMVP | Hbench | POPE | |
| LLaVA | ✗ | Vicuna-v1.5-7B | LLaVA-OV | 57.2 | 44.1 | 49.6 | 39.2 | 71.7 | 71.7 | 68.7 | 58.2 | 37.9 | 70.7 | 56.7 | 40.0 | 36.1 | 85.5 | 56.2 |
| **ASVR** | ✓ | Vicuna-v1.5-7B | LLaVA-OV | **60.0**(+2.8) | **46.5**(+2.4) | **51.3**(+1.7) | **41.7**(+2.5) | **72.2**(+0.5) | **73.2**(+1.5) | **69.9**(+1.2) | **59.8**(+1.6) | **39.7**(+1.8) | **71.8**(+1.1) | **57.5**(+0.8) | **42.0**(+2.0) | **37.9**(+1.8) | **86.9**(+1.4) | **57.9** |

### A.5 THE IMPACT OF VISUAL SUPERVISION

We further extend our experiments by employing discrete SigLIP2 (Tschannen et al., 2025) as visual supervision, which provides richer and stronger semantic information, to verify that enhanced visual-semantic supervision can better scale the effectiveness of ASVR. To ensure fair comparison, we use LLaVA-Next (Liu et al., 2024b) as the training dataset under identical conditions, evaluating ASVR against the baseline with SigLIP (Zhai et al., 2023) as both visual input and supervision, as well as with SigLIP2 (Tschannen et al., 2025) serving the same roles.

The results shown in Table 10 clearly demonstrate that stronger visual semantic encoders lead to better performance when used for supervision. Specifically, ASVR with SigLIP-2 outperforms the baseline (LLaVA) with SigLIP-2 by an average of +2.2 points across 14 benchmarks. In comparison, ASVR with SigLIP improves over its baseline by +1.3 points. These results indicate that ASVR benefits more from stronger semantic supervision, and that pairing ASVR with more powerful semantic vision supervision further enhances its ability to improve visual understanding.

Table 10: **Extend experiments on LLaVA-Next dataset, LLaVA indicates the baseline (typically LVLM framework), ASVR builds upon the baseline by introducing autoregressive semantic visual supervision.** "✗" indicates the use of textual supervision only. Visual encoder(SigLIP-ViT-SO400M/14@384 and SigLIP2-ViT-SO400M/14@384) are both utilized for ASVR and baseline to get different visual input and visual supervision.

| | Visual Encoder | Visual Supervision | LLM Backbone | Data | TVQA | DVQA | OCRB | CQA | MMB | MME | SEED | GQA | MMMU | AI2D | RQA | MMVP | Hbench | POPE | AVG |
|---|---|---|---|---|---|---|---|---|---|---|---|---|---|---|---|---|---|---|---|
| LLaVA | Siglip-so400m-384 | ✗ | Vicuna-v1.5-7B | LLaVA–Next | 57.7 | 40.7 | 37.9 | 42.6 | 67.4 | 71.5 | 67.2 | 61.8 | 34.3 | 65.3 | 54.6 | 32.8 | 33.1 | 86.4 | 53.8 |
| ASVR | Siglip-so400m-384 | Semantic Siglip | Vicuna-v1.5-7B | LLaVA–Next | 58.6 | 40.6 | 39.7 | 43.4 | 67.9 | 73.0 | 67.5 | 62.9 | 34.2 | 65.8 | 55.4 | 36.8 | **39.2** | 85.9 | 55.1 |
| LLaVA | Siglip2-so400m-384 | ✗ | Vicuna-v1.5-7B | LLaVA–Next | 59.2 | 41.8 | 40.5 | 46.3 | 66.9 | 74.0 | 68.3 | 62.7 | 34.7 | 66.4 | **56.9** | 33.3 | 35.1 | 86.1 | 55.2 |
| ASVR | Siglip2-so400m-384 | Semantic Siglip-2 | Vicuna-v1.5-7B | LLaVA–Next | **61.0** | **43.7** | **44.8** | **49.9** | **70.2** | **76.8** | **69.5** | **63.4** | **36.3** | **67.3** | 56.7 | **42.0** | 35.8 | **86.8** | **57.4** |

### A.6 COMPARISON WITH ROSS

Specifically, ROSS (Wang et al., 2024b) reconstructs continuous, appearance-level visual features (VAE features) through denoising, whereas our ASVR reconstructs discrete, semantic-level visual indices (such as discretized SigLIP features) via autoregression. we conducted experiments using the LLaVA-Next dataset (Liu et al., 2024b) under identical training settings, clearly demonstrating that the ASVR-trained model consistently outperforms the ROSS ablation variants across multiple multimodal evaluation metrics, we also implemented an additional variant of ROSS that reconstructs continuous semantic-level features (SigLIP features) through denoising. The result is shown in the table below shown in Table 11.

Our ASVR-trained model still achieved the best performance, indicating that autoregressive semantic visual reconstruction (ASVR) is superior to both denoising semantic visual reconstruction ablation variants and even denoising appearance visual reconstruction (ROSS) ablation variants. Reconstructing semantic-level visual information markedly surpasses reconstructing appearance-level

details, and the unified autoregressive modeling paradigm consistently outperforms its denoising-based counterpart. In conclusion, our ASVR approach delivers superior visual reconstruction and modeling compared with the ROSS method, achieving better performance under identical training conditions. We attribute this performance gap to a fundamental alignment principle: large language models (LLMs) are inherently trained to model high-level semantic information. Therefore, when the visual supervision is semantically aligned with textual inputs—as in ASVR—it naturally leads to better integration and understanding. In contrast, reconstructing low-level visual features (as in appearance-based ROSS) lacks semantic alignment and can even hinder comprehension. Since tasks such as VQA rely heavily on semantic reasoning, reconstructing semantic visual information is more effective for enhancing multimodal understanding.

Table 11: **The detailed comparision between ASVR and ROSS ablation variant, ASVR achieve the best performance under identical training conditions.** ROSS models visual information through a denoising approach, whereas ASVR adopts unified autoregressive paradigm. The SigLIP-ViT-SO400M/14@384 is utilized for semantic visual supervision and VAE features is appearance visual supervision.

| Method | Visual Supervision | LLM backbone | Visual Modeling | Data | TVQA | DVQA | OCRB | CQA | MMB | MME | SEED | GQA | MMMU | AI2D | RQA | MMVP | Hbench | POPE | AVG |
|---|---|---|---|---|---|---|---|---|---|---|---|---|---|---|---|---|---|---|---|
| ROSS | Apperance VAE | Vicuna-v1.5–7B | Denoising | LLaVA–Next | 56.3 | 39.6 | 35.9 | 41.0 | 65.6 | 71.7 | 65.9 | 61.6 | 34.4 | 65.5 | 55.0 | 33.3 | 28.9 | 85.9 | 52.9 |
| ROSS | Semantic Siglip | Vicuna-v1.5–7B | Denoising | LLaVA–Next | 57.5 | 40.2 | 37.4 | 42.5 | 67.0 | 70.5 | 66.2 | 62.1 | **34.9** | 64.6 | **55.7** | 30.1 | 31.2 | 85.8 | 53.3 |
| ASVR | Semantic Siglip | Vicuna-v1.5–7B | Autoregressive | LLaVA–Next | **58.6** | **40.6** | **39.7** | **43.4** | **67.9** | **73.0** | **67.5** | **62.9** | 34.2 | **65.8** | 55.4 | **36.8** | **39.2** | **85.9** | **55.1** |

