# OpenReview forum: "Autoregressive Semantic Visual Reconstruction Helps VLMs Understand Better"
_ICLR.cc/2026/Conference — ICLR 2026 Conference Withdrawn Submission_

### Official Review · Reviewer_1LQ4 · 2025-10-29

**Soundness:** 3
**Presentation:** 3
**Contribution:** 3
**Rating:** 6
**Confidence:** 3

**Summary:**

The paper introduces Autoregressive Semantic Visual Reconstruction, an auxiliary objective for large vision–language models that trains them to predict discrete semantic tokens derived from images, in addition to standard text next-token prediction. By replacing pixel-level reconstruction with semantic token prediction, ASVR aims to unify vision and language learning under a shared autoregressive framework. Implemented as a lightweight visual head, the method integrates seamlessly into LLaVA-style architectures and shows consistent improvements across multiple benchmarks and resolutions, suggesting that semantic-level supervision strengthens visual grounding and multimodal understanding.

**Strengths:**

- **Timely and relevant contribution.** The paper addresses a key open problem in multimodal learning: how to achieve genuine, integrated visual–language understanding in large models.
- **Practical plug-in for existing LVLMs.** ASVR can be added to standard LLaVA-style training pipelines with minimal engineering effort and yields consistent, if modest, gains across benchmarks and resolutions.

**Weaknesses:**

- The introduction emphasizes ASVR as a mechanism to leverage unlabeled or image-only data, arguing that autoregressive semantic reconstruction can replace textual supervision when captions are missing. Yet all reported experiments, including both pretraining and instruction-tuning, rely exclusively on captioned or instruction datasets (LLaVA-1.5, Bunny, LLaVA-OV, etc.), and no experiment incorporates genuinely unlabeled images. Consequently, the central motivation is untested: the method’s benefits might arise solely from richer supervision on existing paired data, not from the promised ability to exploit unannotated images. A small-scale study mixing image-only batches with the ASVR loss would be required to substantiate this claim.
- The method introduces a new "Visual Head" module, increasing total model size and memory. This addition, along with the auxiliary vision loss computation at every step and the need to pre-compute semantic tokens for the entire dataset, creates a notable training and storage overhead that could limit its practical adoption.

**Questions:**

- **Clarification needed on autoregressive implementation.** Equation (3) shows L^vision_AR with conditioning p(x^I_i | x^I_{<i}), suggesting autoregressive generation. However, the paper does not specify whether: (1) ground-truth tokens x^I_{<i} are fed as discrete inputs during training (teacher forcing), or (2) all positions are predicted in parallel from continuous features H_I. If (2), the method is non-autoregressive despite the notation, and gains may stem from semantic token prediction rather than sequential structure. Could the authors clarify the exact conditioning and, if non-autoregressive, explain why the autoregressive framing is maintained throughout?

---

### Official Review · Reviewer_VWzq · 2025-10-31

**Soundness:** 2
**Presentation:** 3
**Contribution:** 2
**Rating:** 4
**Confidence:** 3

**Summary:**

Autoregressive Semantic Visual Reconstruction (ASVR) augments large vision–language models (LVLMs) with autoregressive supervision over visual semantics alongside text. Rather than predicting raw pixels, ASVR trains a single autoregressive objective to reconstruct discrete, high-level semantic tokens extracted from images, unifying visual and textual learning within one framework.

Contributions
1. Framework. Addresses the limits of text-only supervision by explicitly modeling visual semantics autoregressively.
2. Effectiveness. Across 14 benchmarks, ASVR yields consistent gains (e.g., +3.0% average for LLaVA-1.5), while appearance-based reconstruction degrades performance.
3. Scalability. Benefits persist across data scales, model architectures, and high-resolution settings, indicating strong robustness.

**Strengths:**

1. Originality. ASVR proposes a new paradigm for visual supervision in LVLMs by training on discrete semantic tokens (e.g., via DualToken) within an autoregressive objective—departing from pixel-level or diffusion-based reconstruction and moving beyond text-only pretraining.
2. Quality. The evaluation is comprehensive: 14 benchmarks, multiple backbones (Vicuna-7B/13B, Mistral-7B), and data scales from 665K to 3.5M samples. Ablations are convincing; for example, dual-stage training yields ~6% gains (Table 4).
3. Clarity. The paper is accessible, with precise definitions of key terms and clear figures (e.g., attention maps in Fig. 3) that explain the training pipeline and behavior of semantic reconstruction.
4. Significance. ASVR scales across data regimes and architectures, and adapts cleanly to high-resolution settings (Table 3), delivering consistent improvements in multimodal understanding without incurring massive data overhead.

**Weaknesses:**

1. Baseline coverage. Comparisons beyond LLaVA and ROSS are needed. Include strong, recent LVLMs (e.g., Qwen-VL, InternVL) under matched settings to substantiate superiority with both zero-shot and fine-tuned results.
2. Operational costs. The compute overhead of semantic tokenization and autoregressive visual decoding is unreported. Quantify training and inference latency, memory footprint, throughput, and FLOPs across resolutions/sequence lengths, with and without tokenizer caching.
3. Tokenizer dependence. Reliance on DualToken/SigLIP may bake in biases. Evaluate alternative tokenizers and freezing vs. finetuning strategies to demonstrate robustness and generality.
4. Theory and insight. Provide a rationale for why semantic reconstruction helps—e.g., an information-theoretic view (mutual information, rate–distortion), representational analyses (CKA/probing), or controlled perceptual/user studies and causal masking ablations to link gains to semantics rather than confounds.

**Questions:**

1. Scaling & objectives. How does ASVR scale with tokenizer vocabulary size and token granularity, and with alternative semantic objectives? Please report scaling curves (accuracy vs. vocab/sequence length) under matched compute.
2. Generality to video/3D. Can ASVR extend to video or 3D? What architectural changes (spatiotemporal tokenizers, causal temporal decoding, point/voxel tokens) and training tweaks (memory buffers, masking) are required, and with what compute implications?
3. Why appearance hurts. What mechanism explains the drop with appearance/pixel reconstruction—optimization conflict, noisy targets, or priors misalignment? Include controlled ablations (loss interpolation, target noise, encoder freezing) and representational probes.
4. Efficiency–performance frontier. What are the trade-offs when adding visual supervision? Provide Pareto curves of accuracy vs. wall-clock/GPU-hours/latency/memory, and assess the impact of token caching and precision settings.

---

### Official Review · Reviewer_17uL · 2025-10-31

**Soundness:** 3
**Presentation:** 3
**Contribution:** 3
**Rating:** 4
**Confidence:** 3

**Summary:**

This paper addresses the issue that typical Large Vision-Language Models (LVLMs) often overlook visual information by focusing autoregressive supervision only on textual sequences. The authors introduce Autoregressive Semantic Visual Reconstruction (ASVR), which applies autoregressive supervision to the visual modality as well. ASVR reconstructs discrete semantic visual tokens, showing that semantic-level supervision is more effective than appearance reconstruction. The method is conceptually simple and experimentally validated across multiple instruction-tuning datasets, evaluation benchmarks, and LLM backbones, demonstrating strong scalability and consistent performance gains.

**Strengths:**

• The paper is well-motivated and provides a clean and intuitive extension of autoregressive supervision from text to vision. ASVR is simple and integrates naturally into existing LVLM architectures.
• Experiments are relatively comprehensive, covering different visual tokenizers, visual encoders, LLM backbones, and a variety of multimodal benchmarks. Results on LLaVA-1.5 show clear improvements.

**Weaknesses:**

• The paper lacks intuitive qualitative examples beyond the attention maps shown in Appendix A.2. For example, visualizations from tasks like TextVQA could better illustrate how ASVR improves visual modeling and grounding.
• The experiments focus mainly on LLaVA-1.5. It would strengthen the paper to evaluate on additional LVLM architectures (e.g., LLaVA-OV or video-based models) to confirm generality. Demonstrating consistent improvements across other model families or modalities (e.g., video) would significantly increase the paper’s credibility and impact.

**Questions:**

• Since vector quantization (VQ) is inherently lossy, how sensitive is ASVR to the visual tokenizer design and codebook size? How much semantic information is lost during quantization?
• ASVR enforces autoregressive reconstruction of all visual tokens. Given that many tokens may be redundant or encode low-level details, is full reconstruction necessary? Could partial or soft reconstruction achieve similar performance more efficiently?
• The experiments are mainly conducted on LLaVA-1.5. Can ASVR generalize to other LVLM architectures (e.g., LLaVA-OV, or video-based LVLMs) as claimed?

---

### Official Review · Reviewer_k4CW · 2025-10-31

**Soundness:** 2
**Presentation:** 2
**Contribution:** 1
**Rating:** 2
**Confidence:** 3

**Summary:**

The paper presents Autoregressive Semantic Visual Reconstruction (ASVR), a training strategy to improve the visual understanding performance of vision-language models (VLMs) by doing next-token prediction on the image patch tokens as well as text tokens. The authors show performance improvements on models in the LLaVA family by training the LLM backbone to predict discrete image tokens obtained using a semantic tokenizer.

**Strengths:**

- The paper is for the most part accessible and well-written. The main design choices are clearly presented and motivated.
- The integration between visual and textual data in VLMs is a timely topic, and clearly an open research issue. Methods that allow better information flow from images to text are particularly valuable.

**Weaknesses:**

- **Relation with DualToken [a].** The method presented in this paper has a strong link with the DualToken model, and the authors do not provide sufficient information to correctly position it with respect to such previous work. Specifically, the DualToken paper proposes an approach to train a VLM by applying next-token prediction on image patches, both using semantic and pixel tokens as target labels. This seems extremely related to the ASVR recipe proposed in the present manuscript, hence a clarification regarding the novelty of its contributions is needed.
- **Method generality:** It is unclear whether the ASVR method can be applied outside the exact conditions employed by the authors. Particularly, it is assumed (line 218) that the sequence of image features and discrete visual tokens have the same length. While this is true in the paper setup, as DualToken is based on the image encoder of choice (SigLIP), this condition would not be trivially met with any other choice of tokenizer and/or encoder. Given this apparently strong reliance on DualToken, it would be particularly useful to provide a more detailed description of the tokenizers employed, even more so as the original work [a] is very recent and presently unpublished. For instance, it is currently quite obscure why one would have a visual target of shape 27 × 27 × 8 (line 295) or what a 'residual depth' of 8 means in this context.
- **Unclear motivation:** One of the main motivations provided by the authors for ASVR is the scarcity of text-annotated image datasets. However, it is not evident how the proposed method represents a solution in this direction, as no experiment explicitly shows that ASVR can reduce the reliance of VLMs on paired text-image datasets.

**Reference:**

[a] Song et al., Dualtoken: Towards unifying visual understanding and generation with dual visual vocabularies, arXiv 2025

**Typos**:

- 'multimoda' appears instead of 'multimodal' several times (lines 231, 325, 347, 356, 362, 366, 379, 419)
- 'is is' (line 250)
- 'where utilize' (line 243)
- 'we observe ASVR consistent' missing a verb (line 315)
- 'provide' -> 'providing' (line 358)
- 'provide' -> 'be provided' (line 407)

**Questions:**

- How does the high-resolution pipeline work (section 5.4)? In particular, how can the same SigLIP model used in the previous sections (at resolution 384x384) be used to encode images at resolution 1152x1152?
- What is the 'discrete SigLIP2' encoder employed in section 5.5?

---

### Public Comment · ~Yubo_Zhu3 · 2025-11-21
**concerns about the writing**

I would like to raise a concern regarding the writing and presentation of this submission.

Although the authors cite the ROSS [1] paper, some parts of the manuscript appear to resemble ROSS in structure, narrative style, and specific phrasing. Several portions of the Introduction, Preliminary, and Method sections share wording patterns with those found in ROSS, which may make it somewhat difficult to clearly distinguish the authors’ own explanations from previously published descriptions.

A more specific point appears in Appendix A.2. The caption of Figure 3 in this submission matches the caption of Figure 8 in the ROSS paper, including the sentence: “ROSS urges the model to focus on specific image contents corresponding to the question with higher attention values.” The name “ROSS” remains unchanged here.

I hope the authors can consider revising the manuscript, if appropriate, to more clearly differentiate its writing and presentation from ROSS.

[1] Wang, H., Zheng, A., Zhao, Y., Wang, T, Ge, Z., Zhang, X., & Zhang, Z. (2024). Reconstructive visual instruction tuning. arXiv preprint arXiv:2410.09575.

---

### Note · Authors · 2025-11-22

I have read and agree with the venue's withdrawal policy on behalf of myself and my co-authors.